# A Novel Strategy for Identifying NSCLC MicroRNA Biomarkers and Their Mechanism Analysis Based on a Brand-New CeRNA-Hub-FFL Network

**DOI:** 10.3390/ijms231911303

**Published:** 2022-09-25

**Authors:** Jin Zhang, Renqing Nie, Mengxi Liu, Xiaoyi Zhang

**Affiliations:** Faculty of Environment and Life, Beijing University of Technology, Beijing 100124, China

**Keywords:** non-small cell lung cancer, miRNA biomarker, independent regulation model, regulatory network, mechanism

## Abstract

Finding reliable miRNA markers and revealing their potential mechanisms will play an important role in the diagnosis and treatment of NSCLC. Most existing computational methods for identifying miRNA biomarkers only consider the expression variation of miRNAs or rely heavily on training sets. These deficiencies lead to high false-positive rates. The independent regulatory model is an important complement to traditional models of co-regulation and is more impervious to the dataset. In addition, previous studies of miRNA mechanisms in the development of non-small cell lung cancer (NSCLC) have mostly focused on the post-transcriptional level and did not distinguish between NSCLC subtypes. For the above problems, we improved mainly in two areas: miRNA identification based on both the NOG network and biological functions of miRNA target genes; and the construction of a 4-node directed competitive regulatory network to illustrate the mechanisms. NSCLC was classified as lung adenocarcinoma (LUAD) and lung squamous cell carcinoma (LUSC) in this work. One miRNA biomarker of LUAD (miR-708-5p) and four of LUSC (miR-183-5p, miR-140-5p, miR-766-5p, and miR-766-3p) were obtained. They were validated using literature and external datasets. The ceRNA-hub-FFL involving transcription factors (TFs), microRNAs (miRNAs), mRNAs, and long non-coding RNAs (lncRNAs) was constructed. There were multiple interactions among these components within the net at the transcriptional, post-transcriptional, and protein levels. New regulations were revealed by the network. Meanwhile, the network revealed the reasons for the previous conflicting conclusions on the roles of *CD44*, *ACTB*, and *ITGB1* in NSCLC, and demonstrated the necessity of typing studies on NSCLC. The novel miRNA markers screening method and the 4-node directed competitive ceRNA-hub-FFL network constructed in this work can provide new ideas for screening tumor markers and understanding tumor development mechanisms in depth.

## 1. Introduction

Lung cancer (LC) is one of the primary causes of cancer-related deaths worldwide [1]. Approximately 80–85% of LC is classified as non-small cell lung cancer (NSCLC), and the majority of NSCLC comprises two major histological subtypes: lung adenocarcinoma (LUAD) and lung squamous cell carcinoma (LUSC) [2]. Although significant progress has been made in the diagnosis and treatment strategies for LC, the 60-month overall survival (OS) and 5-year survival rates for LC are only 16% to 18%, because 70% to 80% of LC patients are already at an advanced stage when initially diagnosed [3]. With early diagnosis and treatment, the 5-year survival rate for LC can approach 45–65% [4]. Therefore, it becomes increasingly important to diagnose LC as early as possible.

The diagnosis of LC relies mainly on imaging, cytology, and biochemical examination. Due to the various limitations, there are problems with a high false-positive rate, low sensitivity, and specificity [5]. The use of microRNAs (miRNAs) in diagnosis and prognosis has several advantages: they can be detected in circulating plasma [6]; they have higher stabilities [7]; there are statistically significant differences in miRNA expression levels in the serum of NSCLC patients compared to normal tissues [8]; they are specific in tissue expression and temporal expression to reflect the evolutionary pattern of disease development easily and accurately [9]. Moreover, nearly half of the annotated human miRNAs are located at vulnerable and critical points in the genome [10]. Many studies have shown that miRNA biomarkers have the potential to be used in the diagnosis and prognosis of LC [11,12].

Most studies identifying miRNA biomarkers only consider the changes in the expression level of miRNA itself, but do not involve its regulatory genes. The difference in the expression levels of miRNAs can be reflected in the expression levels of the target genes regulated by miRNAs. The underlying mechanisms of various biological phenomena cannot be illustrated by an individual bio-molecule but a well-studied biological network can reveal the inherent laws of life activities at the systemic level. The systematic computational method of integrating miRNA regulatory data and gene expression data will be more effective in inferring potentially aberrant miRNA activity in cancer [13]. From this point of view, several regulatory models based on miRNA-mRNA regulation networks have been used in biomarker identification works. However, these studies rely heavily on the training set, leading to high false positives. Moreover, in these studies, the miRNA that regulates hub genes in the co-regulation model is regarded as the most critical molecule, and it is chosen as a biomarker [14]. Recently, the miRNA-independent regulation model has been proposed to quantify the possibilities of miRNA biomarkers by two metrics [15]: the number of genes uniquely targeted (NOG) by a specific miRNA and the percentage of transcription factor genes (TFP) targeted by the miRNA. The NOG values reflect the independent regulatory ability of miRNAs, and the TFP values reflect the functional importance of miRNA-regulated genes. The independent regulation model is more “fragile” and “influential” than the co-regulation model. MiRNAs can be used as biomarkers if they can independently regulate hub genes with important roles in the protein–protein interaction (PPI) network [16]. Some investigators have well demonstrated the generality and predictive ability of the independent regulation model in complex diseases, such as prostate cancer [17], colorectal cancer [18], and so on [19]. However, the model only divides regulatory genes into genes that are independently regulated, genes that are non-independently regulated, and transcription factor (TF) genes, but does not consider the biological significance of miRNA-regulated genes. It is well-known that genes often play different roles in different environments. We believe that if the target of a miRNA is associated with a disease, then this miRNA will be more involved in the evolution of the disease. Therefore, the combination of the independent regulatory model with the biological function of miRNA target genes will make the obtained markers more reliable (Figure 1).

Biological regulation is complex and multilayered. The regulatory mechanisms of miRNAs reported before are mostly based on the post-transcriptional level, such as the competition between miRNAs and long non-coding RNAs (lncRNAs) or the interaction between miRNAs and target genes [20], and the obtained conclusions would be very one-sided. By controlling the transcriptional activity of genes at the transcriptional level by binding to cis-regulatory elements of genes [21], TFs can direct cell division, cell growth and death, as well as the migration and organization of cells during embryonic development. TFs are essential for a series of biological processes. The expression of protein-coding genes is regulated at the transcriptional level (TFs) and the post-transcriptional level (miRNAs) [22]. There is growing evidence for cooperation and crosstalk between miRNAs and TFs, primarily to buffer gene expression and/or tune signals [23]. 

In addition, lncRNAs are associated with many biological processes in a variety of diseases, such as cancer and cardiovascular disease [24,25]. LncRNAs can compete with mRNAs for the same miRNA response element (MRE) and alleviate miRNA repression on target genes (mRNAs). Thus, lncRNAs, serving as competing endogenous RNAs (ceRNAs), lead to the down-regulation of these miRNAs in the cytoplasm and have undeniable effects on gene regulation and diseases [26].

In a word, both TFs and lncRNAs affect the development and progression of diseases and are important regulators that cannot be ignored. A complicated network, which combines miRNAs, mRNAs, TFs, and lncRNAs, would deeply reveal the mechanisms of transcriptional and post-transcriptional regulation, while it has rarely been considered before. Moreover, the protein–protein interaction (PPI) network can reveal insights into biological regulatory pathways at the protein level. Investigating underlying biological meaning at the transcript level, post-transcript level to protein level will be more systematic.

In this study, we identify reliable miRNA biomarkers for the diagnosis of the two major subtypes of NSCLC (LUAD and LUSC), respectively. An independent regulatory model including NOG and TFP values of miRNAs and the biological significance of the target genes would be taken into consideration. Furthermore, to understand the potential molecular mechanisms of miRNAs in NSCLC at the systemic level, a 4-node ceRNA-hub-FFL network involving TFs, lncRNAs, genes, and miRNAs would be constructed based on the obtained miRNA biomarkers. In the constructed network, PPI would be used to identify functionally important target genes.

## 2. Results

### 2.1. Data Pre-Processing and Differential Expression Analysis Results

The data of LUAD and LUSC were downloaded from the TCGA database. After data pre-processing, there were 16,498 mRNAs and 523 mature miRNAs for each sample in LUAD, and 16,727 mRNAs and 531 mature miRNAs in LUSC.

Differential expression analysis showed that 1438 mRNAs and 111 miRNAs were up-regulated, and 1084 mRNAs as well as 74 miRNAs were down-regulated in LUAD (Figure 2a,b). In LUSC, there were 1991 up-regulated mRNAs and 1861 down-regulated mRNAs, 125 up-regulated miRNAs, and 83 down-regulated miRNAs (Figure 2c,d).

### 2.2. Screening and Validation of the Lung Cancer Biomarkers

#### 2.2.1. Obtaining miRNA Biomarkers Based on the Independent Regulatory Model

A de-redundant operation was carried out to the data of experimentally validated databases and the computer-predicted databases, so 319,574 and 9405 miRNA-mRNA interaction pairs were left, respectively. The human miRNA-mRNA network was constructed using 325,729 miRNA-mRNA interaction pairs. Then, a disease-specific miRNA-mRNA network was extracted from it using differential expression mRNAs and miRNAs. NOG and TFP values of miRNAs were calculated based on the network. The distribution of NOG values conformed to the Power distribution. Finally, ten candidate miRNAs in LUAD (Appendix A) and nine in LUSC were obtained (Appendix A).

#### 2.2.2. Identifying Candidate miRNA Biomarkers Based on Biological Significance of Target Genes

Obtaining the most relevant gene sets for LC using WGCNA

A WGCNA network was built using mRNA expression data of paired normal and tumor samples of LUAD and LUSC, respectively, and highly cancer-related gene modules were obtained from it (Figure 3). In LUAD, the best soft-thresholding power β = 7 was chosen for constructing the co-expression network based on the fit and connectivity. Eighteen modules were identified using the dynamic cutting algorithm. The correlations between the modules were calculated, and the modules with a correlation bigger than 0.75 were combined into one module, resulting in 13 modules. Normal and tumor samples in LUAD were used as clinical traits, and the strongest negative correlation was detected between the Turquoise module with the tumor samples (r = −0.95, *p* = 1 × 10^−57^). Figure 3a–d shows that the expression profile of genes in the Turquoise module was strongly correlated with LUAD sample traits. Therefore, the gene set in the Turquoise module was selected. Similarly, in LUSC, 12 modules were identified finally, and the strongest negative correlation was detected between the Brown module with the tumor samples (r = −0.96, *p* = 4 × 10^−53^). Figure 3e–h shows that the expression profile of genes in the Brown module was strongly correlated with LUSC sample traits, so the gene set in the Brown module was selected.

Screening LC-related genes based on databases of oncogenes and tumor suppressor genes

In order to determine whether the above-obtained genes are associated with LC, gene-lung-related enrichment analysis was performed using DAVID. A total of 711 genes in LUAD and 1560 genes in LUSC were selected. In addition, further screening was carried out based on whether the genes are oncogenes or tumor suppressor genes. Finally, 100 and 198 oncogenes or tumor suppressor genes were obtained in LUAD and LUSC, respectively (Appendix A).

Identifying miRNA biomarkers based on biological significance of genes

A strongly correlated miRNA-mRNA network was constructed by the above NSCLC-related important genes. The miRNAs in the network were the candidate biomarkers that can regulate important LC-related genes. Finally, 13 candidate miRNAs in LUAD (Appendix A) and 51 candidate miRNAs in LUSC (Appendix A) were obtained.

#### 2.2.3. The Final miRNA Biomarkers for LC

Considering both the independent regulation model and the biological significance of the target genes, two and five candidate biomarkers were obtained in LUAD and LUSC, respectively (Table 1 and Table 2). Then, ROC curves and AUC were used to identify potential biomarkers (AUC > 0.85). The final miRNA biomarker was miR-708-5p in LUAD. There were four miRNAs (miR-766-5p, miR-766-3p, miR-140-5p, and miR-183-5p) in LUSC. The above five miRNAs not only had strong independent regulatory abilities, but their target genes were also biologically important. Those genes are all oncogenes or tumor suppressor genes.

#### 2.2.4. Validating the Reliability and Rationality of miRNA Biomarkers

Validation using literature

Table 3 shows the amount of literature associated with the obtained miRNAs in Web of Science and PubMed databases. The relevance of each miRNA and LC was reported, although the reports are very few. However, it is evident that in most of the literature, subtypes are not distinguished. 

Validation using external datasets

Validation of the ability to distinguish LUAD from normal: Three GEO datasets, GSE102286, GSE63805, and GSE36681, were used to validate the potential to distinguish LUAD from normal based on the expression and AUC of miR-708-5p. Because there were two different sample preparation methods (FF and FFPE) in the GSE36681 dataset, the dataset was bisected. In all three datasets, the expression of miR-708-5p in LUAD was significantly higher than in normal samples, which was consistent with our results (Figure 4). The AUC values of miR-708-5p in the four datasets were 0.70, 0.79, 0.79 and 0.71, respectively. 

Validation of the ability to distinguish LUSC from normal: The GEO datasets GSE15008 and GSE74190 were used to validate the potential to distinguish LUSC from normal based on the expression and AUC of biomarkers. MiR-766-5p could not be validated because there is no miR-766-5p in the sequencing platform microarray. As to miR-140-5p, there was no significant difference in expression in GSE15008 (Figure 5), but it was significantly down-regulated in LC samples than in normal tissues in GSE74190 (Figure 6). The expression levels of other biomarkers were significantly up-regulated in LC samples in the two datasets. The AUC values of miR-140-5p, miR-183-5p, and miR-140-5p were 0.52, 0.85, and 0.61 using GSE15008, respectively, and were 0.96, 0.99, and 0.63 in GSE74190, respectively (Figure 5 and Figure 6). There are no more datasets available for validation in the GEO database.

Validation of the ability to distinguish NSCLC from normal tissue using a TCGA integration dataset without distinction of subtypes: Validation was performed using paired and unpaired samples, respectively. MiR-766-3p was not validated because of too many missing values. In both unpaired and paired samples, miR-183-5p and miR-708-5p were significantly over-expressed, and miR-766-5p and miR-140-5p were significantly low-expressed (Figure 7 and Figure 8). The AUC values of miR-183-5p, miR-708-5p, miR-766-5p, and miR-140-5p were 0.97, 0.68, 0.86, and 0.86 in unpaired samples, and 0.96, 0.65, 0.81, and 0.88 in paired samples, respectively. 

Evaluation of the ability of biomarkers to distinguish between LUAD and LUSC: There are some pathological differences between LUAD and LUSC. However, only a few published papers have explored miRNA biomarkers for stratifying LC subtypes. Thus, we carried out this study.

In terms of expression level, miR-183-5p was not significantly different between the two subtypes. As to miR-766-3p, it could not be compared because of too many missing values in LUAD, but it was detectable in LUSC, indicating that the expression level of miR-766-3p was higher in LUSC than in LUAD. The expression levels of other miRNAs were significantly different between the two subtypes (Figure 9). 

The AUC values of miR-183-5p, miR-708-5p, miR-766-5p, and miR-140-5p were 0.52, 0.59, 0.82, and 0.67, respectively.

### 2.3. Construction of ceRNA-hub-FFL Network Based on miRNA Biomarkers

#### 2.3.1. Construction of FFL Network

MiRNAs and TFs can co-regulate shared target genes in feed-forward loops (FFLs) [27] in at least three situations: (1) TFs regulate miRNA expression in their promoter regions [28], (2) TFs and miRNAs can regulate each other by forming feedback loops (FBLs), and (3) both miRNAs and TFs can co-regulate shared target genes and form feed-forward loops (FFLs). Previous studies have explored the underlying molecular mechanisms of diseases or cellular conditions through the key regulators (miRNAs and TFs) and their interactions in these networks [29].

The interaction pairs of the four regulatory factors, TFs, miRNAs, genes, and lncRNAs, at the transcriptional and post-transcriptional levels, were obtained (Appendix A). Then, TCGA expression data was used to obtain the direction of regulation.

The FFL network was extracted from the above network (Appendix A). There were 4662 I-FFLs, 681 II-FFLs, and 318 III-FFLs (Appendix A).

#### 2.3.2. Obtaining hub-FFL Network and Further Extracting ceRNA-hub-FFL Network

To obtain the key subnets from the above FFL network, a PPI net was constructed (Appendix A) to identify miRNA-regulated target genes whose proteins play important functions, and these key genes were used to extract the hub-FFL subnetworks from the FFL network (Appendix A).

LncRNAs in the cytoplasm can also interact with miRNAs as competitive endogenous RNAs and participate in the regulation of target gene expression, so the subcellular localization of the lncRNAs in the hub-FFL network was analyzed using both the lncLocatorand and iLoc-lncRNA (Figure 10), and the intersection was obtained. Finally, 8 lncRNAs were determined, and they were used to extract the ceRNA-hub-FFL network from the hub-FFL network.

The last ceRNA-hub-FFL network is shown in Figure 11.

### 2.4. Analysis of Potential Molecular Mechanisms of Lung Cancer Based on ceRNA-hub-FFL Network

The ceRNA-hub-FFL subnetwork of each miRNA biomarker was extracted to explore the potential mechanism in depth.

#### 2.4.1. Mechanism Revealed by miR-708-5p Related ceRNA-hub-FFL Regulatory Subnetwork

The miR-708-5p related ceRNA-hub-FFL subnetwork is shown in Figure 12a. As it can be seen, at the transcriptional level, the two TF genes, MAZ and EZH2, act as activators of the miR-708-5p and repressors of the gene *CD44* and the four lncRNAs. At the post-transcriptional level, miR-708-5p acts as a repressor of *CD44*, and the four lncRNAs can competitively combine with miR-708-5p, thus weakening its repressive effect on *CD44*.

According to our differential expression analysis and the ceRNA-hub-FFL subnetwork, it can be inferred that miR-708-5p promoted the development and migration of LUAD by up-regulating itself and further inhibiting *CD44* (Figure 12b). The previous study suggested that the down-regulation of *CD44* (or its variants) was associated with an increased invasive and metastatic capacity [30].

To validate our conclusion, the protein and mRNA expression data of *CD44* in normal tissues versus in LUAD were downloaded from the CPTAC, UALCAN [31], and TIMER [32] databases, respectively, and the data confirmed the down-regulation of *CD44* at the protein and mRNA levels (Figure 12c,d,f). All the above confirmed the reliability of our results.

However, there are opposite views on the role of *CD44* in cancer. These studies found that miR-708-5p directly targeted *CD44* in prostate cancer; that the reduced expression level of miR-708-5p resulted in the increased expression level of *CD44* and *AKT2*, resulting in the initiation, development, and progression of prostate cancer [33]; and that deletion or decrease in *CD44* inhibited cancer stem cell properties, induced cell cycle arrest and apoptosis [34].

To uncover the cause of the two contradictory opinions, we studied the two major subtypes of NSCLC, LUAD, and LUSC, separately. MRNA level data from the UALCAN and TIMER databases showed that *CD44* was significantly down-regulated in LUAD samples compared to normal samples, but the difference was not significant in LUSC (Figure 12d–f). The mRNA data of *CD44* and miRNA data of miR-708 were downloaded from the DepMap (https://depmap.org/portal/, accessed on 19 September 2022) database, and the outliers were removed using z-score (|z-score| < 2). Finally, 55 cell lines of LUAD and 17 of LUSC were obtained. The correlation between miR-708 and *CD44* was calculated. The result showed that, in most cell lines (44/55) of LUAD, miR-708 had a negative correlation with *CD44* (r = −0.5926) (Figure 13a), but no correlation in LUSC (Figure 13b). However, there were a few cell lines (11/55) where the correlation was just the opposite. Due to the extreme lack of data on *CD44* mutation-positive NSCLC cell lines and the precise staging information of these cell lines, the possible reason for the different expression of *CD44* between different cell lines in LUAD needs to be further revealed. The above results showed that the expression of *CD44* was different in the two subtypes. If the two subtypes are not distinguished, completely opposite conclusions may occur depending on the sample sizes of the two subtypes. This further illustrates the necessity of exploring the two subtypes separately.

#### 2.4.2. Mechanism Revealed by miR-183-5p Related ceRNA-hub-FFL Regulatory Subnetworks

There are three types of FFL in the miR-183-5p related ceRNA-hub-FFL subnetworks (Figure 14). As seen from the type II and III networks (Figure 14d,e), mutual inhibition exists between TF ZEB1 and miR-183-5p at the transcriptional and post-transcriptional levels, and the two factors regulate *ITGB1* gene expression simultaneously.

In the TF-meditated FFL (I-FFL) subnetwork (Figure 14a), some TFs activate miR-183-5p and correspondingly repress the two lncRNAs, leading to the suppression of the two genes *ITGB1* and *ACTB*. While other TFs play the opposite roles, they finally activate the two genes. According to our differential expression analysis and the ceRNA-hub-FFL subnetwork, it can be inferred that miR-183-5p was significantly up-regulated in LUSC (Figure 15a), which targeted *ACTB* and ultimately led to the down-regulation of *ACTB* and *ITGB1* in LUSC.

*ACTB* gene encodes β-actin. β-actin is a highly conserved cytoskeletal protein that commonly expresses and is essential for cell migration, mitosis, intracellular transport, and maintenance [35]. It has been considered an endogenous house-keeping gene and reference gene in cells and tissues for many years [36], which leads to its role in cancer being ignored. However, the emerging evidence suggests that *ACTB* expresses unstably and plays a key role in a variety of human diseases, particularly cancer [37]. Previous studies showed that the mRNA level of *ACTB* was down-regulated in esophageal cancer, colon cancer, and LUSC compared to normal tissues [38,39]. The same is also true for *ITGB1*. Some studies indicated that down-regulation of *ITGB1* triggered lung disease and even cancer, such as colon cancer [40] and breast cancer [41]. These may be the mechanisms of miR-183-5p regulating the development of lung cancer.

To further validate the above conclusions, we analyzed the data with different databases at different levels. Using the protein and mRNA expression data of *ACTB* and *ITGB1* from the CPTAC, UALCAN, and TIMER databases, the results showed down-regulated *ACTB* and *ITGB1* in LUSC tissues compared to normal tissues (Figure 15b,c,e,g,h). The mRNA data of *ACTB* and *ITGB1* and miRNA data of miR-183 were downloaded from the DepMap database, and the outliers were removed using z-score (|z-score| < 2). Finally, 55 cell lines of LUAD and 17 of LUSC were obtained. The correlations between miR-183 and *ACTB*, miR-183 and *ITGB1* were calculated, respectively. The results showed that, in all 17 cell lines of LUSC, miR-183 had a negative correlation with *ACTB* (r = −0.5477), and with *ITGB1* (r = −0.5203), respectively, (Figure 16a,c), but no correlation in all 55 cell lines of LUAD (Figure 16b,d). All the above data proved the reliability of the conclusion we obtained.

However, some other studies found the opposite results, suggesting that *ACTB* was up-regulated in some cancers, such as esophageal cancer (ES) and NSCLC (not typed) compared to normal samples [42,43], and a study showed that up-regulated *ACTB* was associated with poor prognosis in LUAD [39]. These studies did not distinguish between LUAD and LUSC. Similarly, different voices had emerged on the role of *ITGB1* in tumors, that is up-regulated *ITGB1* promoted the development of LC [44]. Some studies found that down-regulated *ITGB1* inhibited NSCLC [45], and the samples with up-regulated *ITGB1* in LUAD had worse overall survival [46].

To discern the correctness of the above results, we analyzed the mRNA expression data from the UALCAN and TIMER databases. The results showed that *ACTB* was not significantly different in LUAD in both UCLCAN and TIMER (Figure 15d,g), and *ITGB1* was not significantly different in LUAD in TIMER (Figure 15h). These results further illustrated the necessity of studies for NSCLC subtypes. Unfortunately, *ACTB* has been considered as an endogenous house-keeping gene in cells and tissues and as a reference gene for quantitative experiments for many years [47]. The belief that it expresses stably in cells has led to its neglected role in cancer. Our results brought to light the unstable expression of *ACTB* in NSCLC, making the role of *ACTB* as a cancer reference gene challenging. The mechanism of *ACTB* action in NSCLC also needs to be further elucidated. Similarly, no studies have been seen on miR-183-5p targeting *ITGB1* and thus affecting the development of NSCLC. 

#### 2.4.3. Mechanism Revealed by miR-766-5p Related ceRNA-hub-FFL Regulatory Subnetworks

There are three types of FFL in the miR-766-5p related ceRNA-hub-FFL subnetworks (Figure 17). In the TF-mediated (I-FFL) network, most TFs (right) have repressive effects on miR-766-5p, with activating effects on *CCNB1* and related four lncRNAs, while the other seven TFs act in the opposite way.

In the miR-766-5p-mediated FFL (II-FFL), TFs MAZ, HDAC2, and KDM5B have activating effects on *CCNB1* and lncRNAs. However, miR-766-5p represses the three TFs at the post-transcriptional level. Eventually, these interactions enhance the repressive effect of miR-766-5p on *CCNB1*. According to our differential expression analysis and the ceRNA-hub-FFL subnetwork, it can be inferred that miR-766-5p was significantly down-regulated in LUSC (Figure 18a), and the enhanced inhibitory effect of TFs on miR-766-5p led to the up-regulation of the regulated *CCNB1*. A previous study has confirmed that the protein encoded by *CCNB1* is a regulatory protein involved in mitosis, which has been demonstrated to be associated with the development of LC [48]. Reduced expression of *CCNB1* can induce G2/M cell cycle arrest and promote apoptosis [49]. This may be the mechanism of miR-183-5p regulating the development of lung cancer.

To validate our conclusion, the protein and mRNA expression data of *CCNB1* in normal tissues versus in LUSC and LUAD were analyzed based on the CPTAC, UALCAN, and TIMER databases, respectively, and the data confirmed the significantly up-regulated expression of *CCNB1* (Figure 18b–e). All the above proved the reliability of our results.

#### 2.4.4. Mechanisms Revealed by miR-140-5p Related ceRNA-hub-FFL Regulatory Subnetworks

There are three types of FFL in the MiR-140-5p-related ceRNA-hub-FFL subnetworks (Figure 19).

In the miRNA-mediated FFL (II-FFL) and composite FFL (III-FFL) regulatory networks, TF TRIM28 and miR-140-5p inhibit each other at the transcriptional level and post-transcriptional level. They form a feedback loop that co-regulates the expression of the target genes *CDC6*, *YWHAG*, *CCNB1*, and *BRIC5*. TF MYBL2 participates in the activation of the four target genes at the transcript level also.

Most of the key regulators exist in TF-mediated FFL (I-FFL) (Figure 19a,d). Five genes, *YWHAG*, *YWHAQ*, *CCNB1*, *CDC6*, and *BIRC5*, are regulated by activation or repression of different factors. At the transcriptional level, the TFs ETS1 and GATA1 activate miR-140-5p and correspondingly repress the lncRNA SNHG1, TUG1, and the target gene *YWHAG*. However, the effects of TCF3, KDM5B, EZH2, TRIM28, and HDAC2 are the opposite. As for *YWHAQ*, it is almost the same. Six TFs TCF3, TRIM28, KDM5B, HDAC2, HDAC1, and EZH2 have activating effects on gene *CCNB1* and lncRNAs TUG1 and SNHG1 at the transcriptional level, repressive effects on miR-140-5p. The other five TFs NFIC, JUND, ZEB1, ELF1, and ETS1 have the opposite effect. *CCNB1* is an oncogene. It is possible that six TFs play a dominant role in the repression of miR-140-5p, and thus low-expressed miR-140-5p in LUSC weakens the repressive effect on oncogene *CCNB1*. The same thing is true for *CDC6* and oncogene *BIRC5*. According to our differential expression analysis and the I-FFL, it could be inferred that miR-140-5p was significantly down-regulated in LUSC (Figure 20a), so down-regulated miR-140-5p led to a diminished inhibitory effect on *YWHAG*, *YWHAQ*, *CCNB1*, *BIRC5*, and *CDC6*.

*YWHAG* and *YWHAQ* are members of the 14-3-3 protein family. 14-3-3 proteins have been demonstrated to contribute to the regulation of key cellular processes such as signal transduction, cell cycle control, apoptosis, and malignant transformation [50], especially many cellular processes associated with cancer development [51], and over-expression of *YWHAG* has been observed in a variety of tumors [52]. It was reported that *YWHAG* could inhibit the death of apoptotic cells and promote cell migration in breast cancer [53] and have clinical prognostic significance as an oncogene in advanced NSCLC [54]. *CCNB1* and *BIRC5* are all oncogenes. *BIRC5* is a member of the inhibitors of apoptosis (IAP) gene family, which encodes negative regulatory proteins that prevent apoptotic cell death. Unlike other IAPs, *BIRC5* expresses intensely in most tumors, including LUAD and LUSC [55], and it could serve as a predictive biomarker for NSCLC, especially for LUAD [56], but does not express or only expresses at low levels in most normally differentiated tissues [57], and down-regulated *BIRC5* was positively associated with NSCLC survival time [58]. Cell division cycle 6 (*CDC6*) is an important regulator of DNA replication, and there is evidence that silencing of *CDC6* may lead to G1 phase arrest [59]. Over-expression of *CDC6* has been detected in several types of cancer, and up-regulation of *CDC6* is associated with poor prognosis in cancer patients [60]. A study has confirmed that the expression level of *CDC6* is significantly elevated in NSCLC tumor tissues [61]. This may be the mechanism of miR-140-5p regulating the development of LC.

To further validate the above conclusions, we analyzed the data in different databases at different levels. At the proteomic and mRNA levels of LUSC and LUAD in CPTAC, UALCAN, and TIMER databases, *YWHAG*, *YWHAQ*, *CCNB1*, and *BIRC5* were significantly up-regulated (Figure 18b–e, Figure 20b–j and Figure 21a–c). There is no *CDC6* proteome data in the CPTAC database, but in UALCAN and TIMER databases at the mRNA level, *CDC6* was up-regulated in LUSC and LUAD compared to normal samples (Figure 20k,l and Figure 21d). These all proved the reliability of our findings.

## 3. Discussion

The biology network can deepen our understanding of the intrinsic mechanisms of disease. Based on the network, the complex interactions within the organism can be fully considered from a systematic perspective.

Most studies aiming to explore diagnostic and prognostic miRNA markers are based on the coordinated relationships in the regulatory networks. They look for coordinated relationships in the many-to-many relationships between miRNAs and genes, find disease-related miRNA modules and identify markers from them eventually [17]. Disease-related miRNAs in the caner-miRNA modules enhance the common action on the co-regulated genes, i.e., synergistic effects [62]. However, according to the view of systematic biology [63], biomarkers are important indicators reflecting changes between different biological states, so biomarker identification should consider their effects on system stability. Therefore, a different model—an independent regulation model of miRNAs has been proposed. Independent regulation relationships are fragile. If the protein coded by the miRNA target gene is a key node in the network, the abnormal expression of the miRNA with independently regulatory ability to this gene will lead to large changes in this “fragile” structure and affect the stability of the biological system. In biological networks, the independent regulation model is an important complement to a synergistic effect and is more impervious to the dataset [15]. There are some meaningful studies based on this model [16,64], but this model does not consider the biological functions of the target genes. We argue that if the miRNA target gene is highly correlated with the disease, then the miRNA will be more directly and definitively involved in the progression of this disease. Therefore, this miRNA is more likely to become a marker for the disease. The identification of markers using a low-cost and highly accurate method is only the first step. It is more important to gain insight into the potential mechanisms of these markers in disease, which will provide guidance for finding new markers, therapeutic targets, and treatment regimens.

The involvement of miRNAs in the pathogenesis of NSCLC is extremely complex. Most previous studies have analyzed their regulatory mechanisms based only on the post-transcriptional level [65]. This would lead to very one-sided conclusions and even produce erroneous results. The regulation of eukaryotic gene expression can occur at many different hierarchies, including the genetic level, transcriptional level, and post-transcriptional level. Each hierarchy involves multiple factors. At the transcriptional level, TFs as important regulatory molecules can directly control the timing, location, and intensity of gene expression. They control the transcription of DNA into mRNA by combining with specific DNA sequences, and they can also bind to RNA produced by gene transcription and then control the transcription, localization, and stability of RNA. At the same time, many TFs are regulated by factors in the signal transduction pathway. LncRNAs are involved in regulating protein-coding genes at multi-levels, including epigenetic regulation, transcriptional regulation, and post-transcriptional regulation. LncRNAs and miRNAs can interact with each other by competing with some shared RNAs to participate in the expression regulation of target genes in the cytoplasm [66].

The above shows the importance and complexity of TFs and lncRNAs in biological regulatory networks. In short, TFs can regulate the expression of mRNA, miRNA, and lncRNA genes at the transcriptional level, and miRNAs can regulate mRNAs (TF mRNAs and non-TF mRNAs) and lncRNAs at the post-transcriptional level. Such a relationship between TFs and miRNAs can connect the transcriptional level and the post-transcriptional level [67]. Based on this bridging role of TFs, we constructed a 4-node directed competitive ceRNA-hub-FFL network based on miRNA markers in NSCLC for the first time, and the regulatory directions were determined in the network based on the correlation between expressions. Based on the constructed ceRNA-hub-FFL network, the mechanisms of the obtained markers were deeply explored, and we indeed got some novel conclusions and depicted a more comprehensive regulatory mechanism. The constructed network revealed for the first time that miR-708-5p affects the development of LUAD by regulating *CD44*, miR-183-5p affects the development of LUSC through regulating *ITGB1* and *ACTB*, and miR-766-5p affects the development of LUSC through regulating *CCNB1*. These conclusions were validated using multiple databases. In addition, the constructed ceRNA-hub-FFL network provided a more comprehensive landscape of the different roles of lncRNAs and TFs in miRNA regulation of gene expression. Meanwhile, through the ceRNA-hub-FFL network, we clarified that the expression levels of *CD44*, *ACTB*, and *ITGB1* are significantly different in LUAD and in LUSC. This is because multiple factors play different regulatory roles in gene expression, and the final results depend on the synergistic effect of various regulatory factors. The necessity of subtyping studies on NSCLC becomes evident.

The network can give us a more comprehensive and systematic understanding of the underlying molecular mechanisms of NSCLC. The constructed 4-node network can be deeply mined at the level of key subnetwork modules, FFL modules, and relations among TFs, miRNAs, genes, and lncRNAs, thus, the inherent laws of NSCLC can be explored at the systemic level. It is expected to find new regulatory pathways and rules from multiple angles and multiple layers, and it is also expected to identify potential oncogenes and drug targets of NSCLC [68,69].

However, another non-coding RNA, circular RNAs (circRNAs), can also act as a sponge to compete with miRNAs to bind mRNAs and indirectly regulate the expression of target genes [70]. A previous study has demonstrated the regulatory function of circRNAs in tumor progression [71]. Therefore, our study is still deficient. The independent regulation model only reflects the regulatory relationship at the miRNA-mRNA level, and it would be more reasonable and valuable if the regulatory effects of lncRNAs and circRNAs on miRNAs and mRNAs are further considered, as well as to further examine the generality of this method. Furthermore, many factors in organisms are temporally and spatially regulated, so a multi-node multi-level dynamic network will reflect the mechanisms more accurately and fully.

## 4. Methods and Materials

### 4.1. Data Download and Pre-Processing

The sequencing data of miRNAs, mRNAs, and lncRNAs in LUAD and LUSC was downloaded from The Cancer Genome Atlas (TCGA, https://portal.gdc.cancer.gov/, accedded on 17 June 2021) and pre-processed. Three datasets (GSE102286, GSE63805, and GSE36681) and another two datasets (GSE74190 and GSE15008) were downloaded from Gene Expression Omnibus (GEO, https://www.ncbi.nlm.nih.gov/geo/, accessed on 18 September 2021) as external validation datasets for LUAD and LUSC biomarkers, respectively.

### 4.2. Differential Expression Analysis of mRNAs and miRNAs

Differentially expressed mRNAs and miRNAs were analyzed using the R package “DESeq2” [72]. The data were normalized using variance stabilizing transformation (VST) and library composition. The threshold for the identification of differentially expressed miRNAs and genes was |log2 (FC)| ≥ 1.5 and adjusted *p*-value ≤ 0.05.

### 4.3. Screening and Validation of miRNA as Biomarkers 

#### 4.3.1. Identification of miRNA Biomarkers Based on the Independent Regulation Model

Construction of human miRNA-mRNA regulatory network

Six databases were used to construct the human miRNA-mRNA regulatory network. Among them, the data in the four databases, miR2Disease [73] (http://www.mir2disease.org/7, accessed on 17 June 2021), miRecords [74] (http://mirecords.biolead.org/, accessed on 17 June 2021), miRTarBase [75] (https://miRTarBase.cuhk.edu.cn/, accessed on 17 June 2021), and TarBase [76] (http://www.microrna.gr/tarbase, accessed on 17 June 2021), mainly comes from biological experiments. The data of the other two databases, miRDB [77] (http://mirdb.org, accessed on 17 June 2021) and miRWalk [78] (http://mirwalk.umm.uni-heidelberg.de, accessed on 17 June 2021), mainly comes from computer algorithm prediction.

Obtaining candidate biomarkers based on miRNA independent regulation model

Differential miRNAs and mRNAs were used to extract disease-specific miRNA-mRNA networks from the human miRNA-mRNA network, and the NOG and TFP values were calculated based on the net. MiRNAs with significantly high NOG and TFP values were screened out with Wilcoxon signed rank test (threshold of NOG and TFP: *p* < 0.05). Because miRNAs with higher NOG values are more likely to be biomarkers [15], the median NOG values of the obtained miRNAs were calculated further, and miRNAs with NOG values bigger than the median were selected as candidate biomarkers.

#### 4.3.2. Screening miRNA Biomarkers Based on the Biological Significance of Target Genes

A weighted gene co-expression network analysis (WGCNA) was built, and the key target gene set was obtained from it. Then, key LC-related genes, oncogenes, and tumor suppressor genes were screened out from these target genes enriched in lung tissue. The miRNA-mRNA target pairs of these key lung cancer-related genes were identified from the interactions obtained experimentally. The Pearson correlation coefficients between the target pairs were calculated, and the interaction pairs with high correlation were selected as the strongly correlated pairs. Finally, miRNAs in strongly correlated pairs were regarded as candidate biomarkers, which regulate key LC-related genes.

Obtaining the gene set most relevant to LC using WGCNA

To obtain the gene modules highly relevant to cancer, mRNA expression data of paired normal and tumor samples of LUAD and LUSC was obtained from the TCGA database, respectively. The top 75% of variant genes were selected by a robust method of median absolute deviation (MAD) [79]. A WGCNA was constructed and Module Eigengene (ME) dissimilarities were calculated using the R package. The association between ME and LC sample phenotypes was assessed by Pearson correlation values. The *p*-value < 0.05 is significant. The module that was highly correlated with the LC sample phenotypes was selected as the gene module most relevant to LC, and it would be used for subsequent analysis.

Screening LC-related genes based on databases of oncogenes and tumor suppressor genes

To screen out the genes expressed in lung tissue from the gene set obtained above, enrichment analysis was performed using DAVID [80] (https://david.ncifcrf.gov/, accessed on 20 June 2021). The lung-enriched genes were further screened in the oncogene database TSGene [81] and tumor suppressor genes database on Gene [82]. Finally, oncogenes or tumor suppressor genes that express in lung tissue and are highly associated with LC were obtained.

Construction of the strong relationship pairs between LC-related genes and miRNAs

To identify the miRNAs that regulate the above-obtained oncogenes or tumor suppressor genes, strong interactions between miRNAs and target genes must be found. The miRNA-mRNA pairs from the four experimentally validated databases (in “Construction of human miRNA-mRNA regulatory network”) were selected and Pearson correlation coefficients between the target pairs were calculated based on TCGA expression data. The threshold of strong relationship pairs: r < −0.3, *p*-value < 0.05.

#### 4.3.3. The Final miRNA Biomarkers for LC

Candidate miRNA biomarkers were obtained from the intersection of the two methods. Then the final biomarkers were identified by the ROC curve (AUC > 0.85).

#### 4.3.4. Validating the Reliability and Rationality of Biomarkers

Validation based on the existing literature

The biomarkers were submitted to the PubMed database. Literature screening criteria for a certain miRNA: “miR-XXX” [All Fields] AND “humans” [MeSH Terms]. Literature screening criteria for whether miRNA is associated with cancer: “Neoplasms” [MeSH Terms] AND “miR-XXX” [All Fields] AND “humans” [MeSH Terms]. Literature screening criteria for the association of a miRNA with LC: “Lung Neoplasms” [MeSH Terms] AND “miR-XXX” [All Fields] AND “humans” [MeSH Terms]. The literature related to NSCLC was searched in the Web of Science database. Literature screening criteria: TS = ((“LUAD” OR “LUSC” OR “NSCLC” OR “non-small cell lung cancer” OR “non-small-cell lung cancer” OR “lung squamous cell carcinomas” OR “lung adenocarcinoma” OR “non-small cell lung carcinomas”) AND (“miR-XX” OR “miR XX” OR “microRNA XX” OR “microRNA-XX”)). The screening covered literature published up to 16 September 2021.

Validation based on external datasets

The ability to distinguish LUAD/LUSC from normal was validated using the expression and ROC of external GEO datasets GSE102286, GSE63805, GSE36681, and GSE74190, GSE15008, respectively. The ability to distinguish NSCLC from normal tissue of the biomarkers was verified using the unpaired and paired data of normal and tumor samples from mixed subtypes in the TCGA database. The ability to distinguish LUAD from LUSC was tested using the expression data and ROC of LUAD and LUSC tumor samples in the TCGA database. *p*-value < 0.05 was considered statistically significant.

### 4.4. Construction of 4-Node ceRNA-hub-FFL Network Based on miRNA Biomarkers

#### 4.4.1. Identifying the Interaction Pairs between TFs, miRNAs, Genes, and lncRNAs and Calculating the Direction of Interactions

First, a TF dataset based on existing databases was established: TFs with a quality rating reaching C and above were downloaded from HOCOMOCO V11 database [83]; TFs were downloaded from HumanTFDB (hust.edu.cn, accessed on 9 October 2021), a sub-database of AnimalTFDB 3.0 [84]; TFs were extracted from the following databases: TRRUST [85], ENCODE [86], TRANSFAC [87], ChIP-X Enrichment Analysis [88], TransmiR [89], and HTRIdb [90]. Finally, the above TFs were merged as the TF dataset for this study. 

The miRNA-TF relationship pairs were obtained based on the TF dataset constructed above. Then, the interaction pairs between the miRNA biomarkers and genes were extracted from the human miRNA-mRNA network constructed in “Construction of human miRNA-mRNA regulatory network”; miRNA-lncRNA relationship pairs were obtained from the intersection of starBase 3.0 database [91] and lncBase database [92]. The interaction pairs of TFs and targets, including TF-miRNA, TF-gene, and TF-lncRNA, were extracted from ENCODE, CHIP-X, HTRIdb, TRANSFAC, and TRRUST databases. Furthermore, TF-miRNA pairs were additionally extracted from puTmiR database [93].

Correlations and interaction directions between all interaction pairs were calculated based on TCGA expression data, and significantly correlated interaction pairs were retained. For miRNA target pairs: Pearson correlation coefficient r ≤ −0.15, *p* < 0.05, for TFs target pairs: Pearson correlation coefficient |r| ≥ 0.3, *p* < 0.05.

To reduce false positives, the intersection of the relationship pairs obtained from the databases and the expression was chosen as the final relationship pair.

#### 4.4.2. Construction of FFL Network

After obtaining the interaction pairs between the four kinds of factors, three types of 4-node FFL (feed-forward loop) subnetworks were extracted using MotifPredictor (publicly available at https://www.uth.edu/bioinfo/software.htm and https://github.com/emanlee/MotifPredictor, accessed on 9 December 2021). The three types are TF-mediated FFL (I-FFL), miRNA-mediated FFL (II-FFL), and composite FFL (III-FFL), respectively (Figure 22). 

#### 4.4.3. Obtaining hub-FFL Network

Cytohubba’s “degree” and MCC methods were used to extract the top 5% of nodes as the key nodes to construct the FFL key subnetworks [94]. Further, considering the importance of the genes co-regulated by TFs and miRNAs, the protein products of miRNA-regulated genes were extracted to construct a PPI network. The genes of the top 10% of nodes extracted from the PPI network were used to extract the hub-FFL network from the FFL key subnetworks.

#### 4.4.4. Extraction of ceRNA-hub-FFL Network

For the lncRNAs in the hub-FFL networks, LNCipedia (https://lncipedia.org/, accessecd on 13 January 2022) was used to obtain their base sequences. Then, two different algorithms were used in the databases lncLocator 2.0 (http://www.csbio.sjtu.edu.cn/bioinf/lncLocator/, accessecd on 13 January 2022) and iLoc-lncRNA (2.0, http://lin-group.cn/server/iLoc-LncRNA(2.0)/home.php, accessecd on 13 January 2022) to predict the subcellular localization of lncRNAs. LncRNAs localized in the cytoplasm were retained according to the intersection of the two algorithms. Lastly, the ceRNA-hub-FFL network was extracted from the hub-FFL network using obtained lncRNAs.

## 5. Conclusions

A novel miRNA markers screening method and a 4-node directed competitive ceRNA-hub-FFL network were constructed in this work. Finally, miRNA biomarkers of LUAD and LUSC were obtained and validated, respectively, and the new target genes regulated by these biomarkers were revealed to be associated with LUAD and LUSC based on the constructed ceRNA-hub-FFL network. Moreover, we clarified that the expression levels of these target genes are significantly different in LUAD and in LUSC.

## Figures and Tables

**Figure 1 ijms-23-11303-f001:**
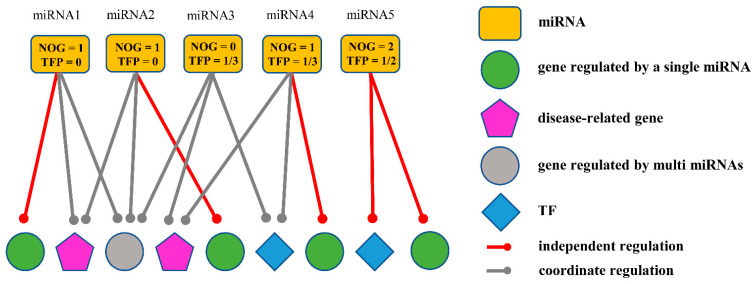
MiRNA-mRNA regulatory network combines the independent regulatory model with the biological function of target genes. MiRNA4 is more likely to be a marker than other miRNAs in the figure because its targets include independent genes, TF genes, and important disease-related genes.

**Figure 2 ijms-23-11303-f002:**
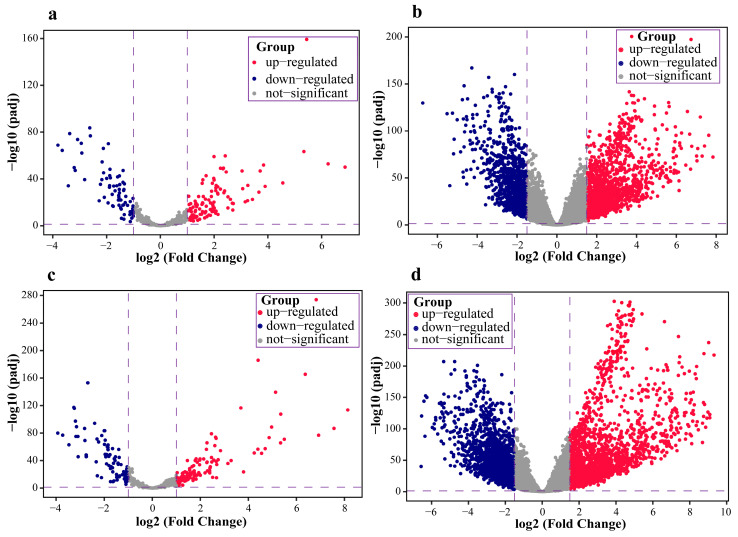
Volcano plot of miRNAs and mRNAs in LUAD and LUSC datasets: (**a**,**b**) indicate the volcano plot distribution of miRNAs and mRNAs in LUAD, respectively; while (**c**,**d**) indicate the volcano plot distribution of miRNAs and mRNAs in LUSC, respectively.

**Figure 3 ijms-23-11303-f003:**
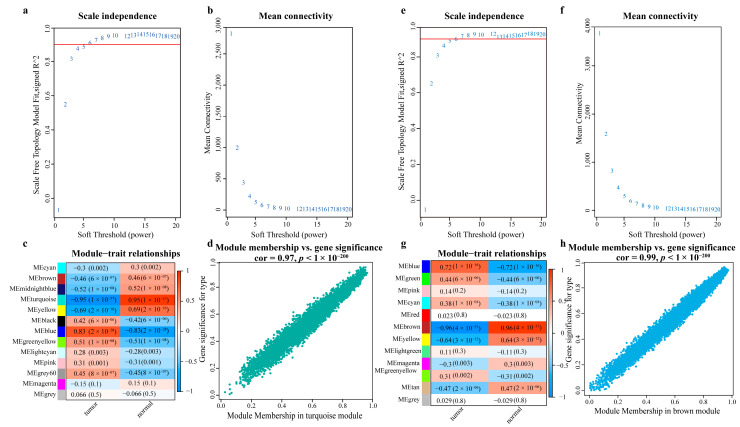
Identification of LUAD/LUSC-related gene set based on WGCNA. (**a**,**e**) are the scale-free fit indices of various soft-thresholding powers, both with the best soft-thresholding value of 7. (**b**,**f**) The average connectivity of various soft-thresholding powers, and the y-axis is a decreasing function of the soft-thresholding power β(x-axis). (**c**,**g**) The gene modules associated with the clinical features of LUAD/LUSC. Each module contains the corresponding correlation and *p*-value, and the correlation coefficient represents the correlation between the gene modules and the clinical features; the color shade represents the correlation size, red indicates a positive correlation, and blue indicates a negative correlation. (**d**,**h**) Scatter plots of the clinical trait between module membership (x-axis) and gene significance (y-axis) in the Turquoise module. Module membership (x-axis) refers to the correlation between the genes and the module, with larger values indicating the greater correlation between the genes and the module, and gene significance refers to the correlation between the genes and the trait, with larger values indicating the stronger correlation between the genes and the trait (LUAD: r = 0.97, *p* < 1× 10^−200^; LUSC: r = 0.99, *p* < 1× 10^−200^).

**Figure 4 ijms-23-11303-f004:**
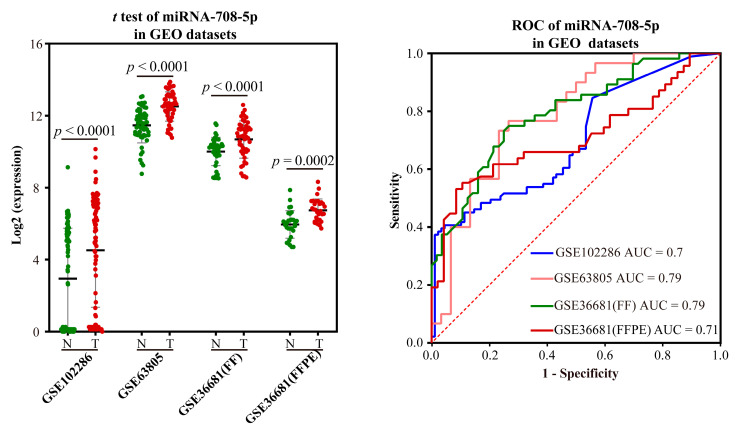
Validation results of miRNA markers in LUAD. FF: Fresh Frozen; FFPE: Formalin-Fixed and Paraffin-embedded. The green dot indicates the normal sample, while the red dot indicates the tumor sample.

**Figure 5 ijms-23-11303-f005:**
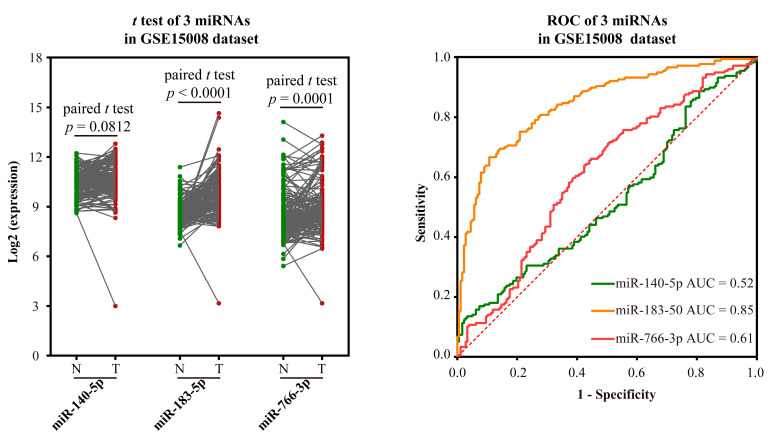
Validation results of 3 miRNA markers in LUSC in GSE15008 dataset. The green dot indicates the normal sample, while the red dot indicates the tumor sample, and the line indicates the paired samples.

**Figure 6 ijms-23-11303-f006:**
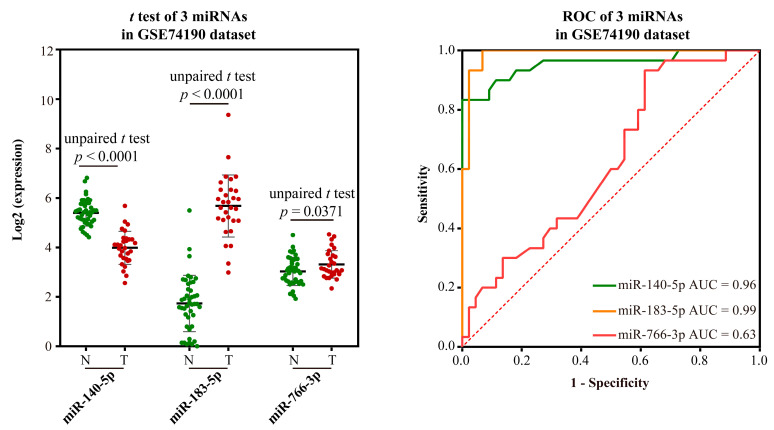
Validation results of 3 miRNA markers in LUSC in GSE74190 dataset. The green dot indicates the normal sample, while the red dot indicates the tumor sample.

**Figure 7 ijms-23-11303-f007:**
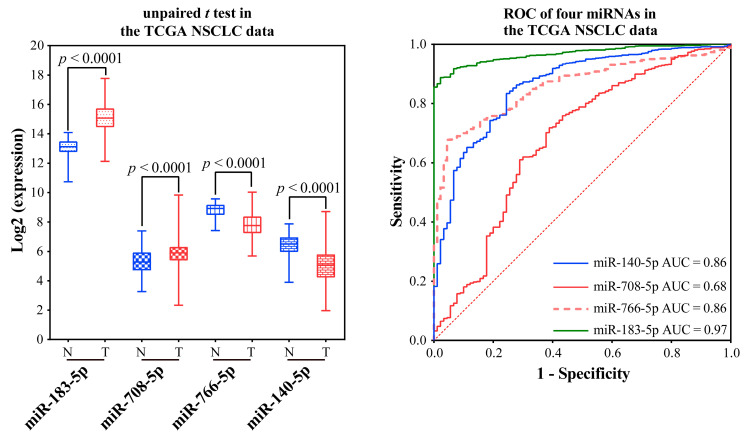
Validation of miRNA markers in unpaired NSCLC and normal samples in TCGA. The blue box indicates the normal samples, while the red box indicates the tumor samples.

**Figure 8 ijms-23-11303-f008:**
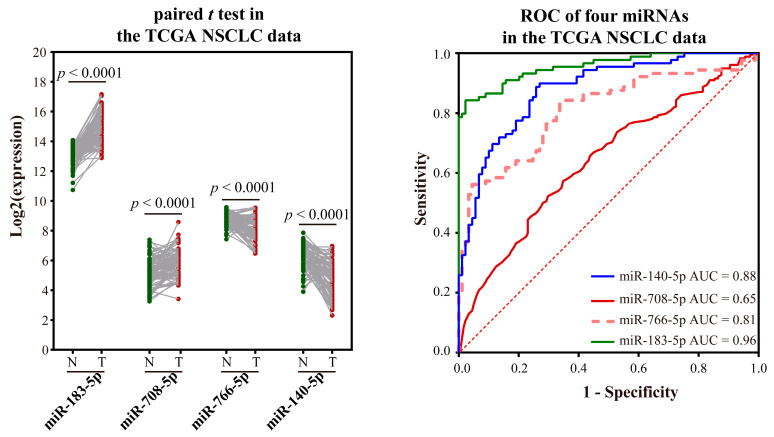
Validation of miRNA markers in paired NSCLC and normal samples in TCGA. The green dot indicates the normal sample, while the red dot indicates the tumor sample, and the line indicates the paired samples.

**Figure 9 ijms-23-11303-f009:**
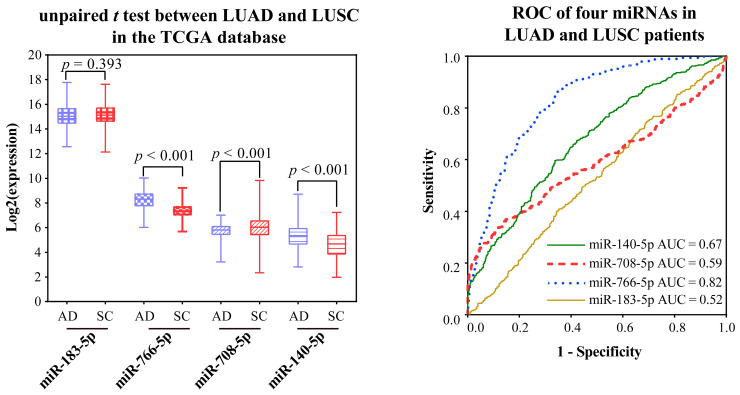
Expression and ROC curves of miRNA markers in LUAD and LUSC. The blue box indicates the normal samples, while the red box indicates the tumor samples.

**Figure 10 ijms-23-11303-f010:**
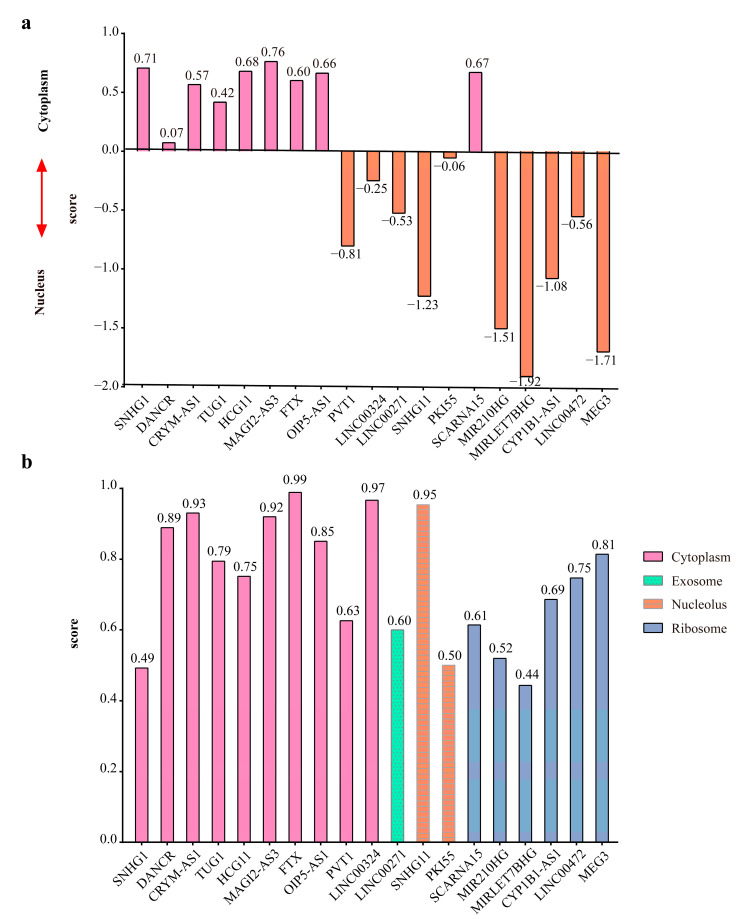
Subcellular localization of lncRNAs: (**a**) Subcellular localization of lncRNAs in hub-FFL network predicted in lncLocator based on deep learning of A549 cell line data; (**b**) Subcellular localization of lncRNAs in hub-FFL network predicted in iLoc-lncRNA.

**Figure 11 ijms-23-11303-f011:**
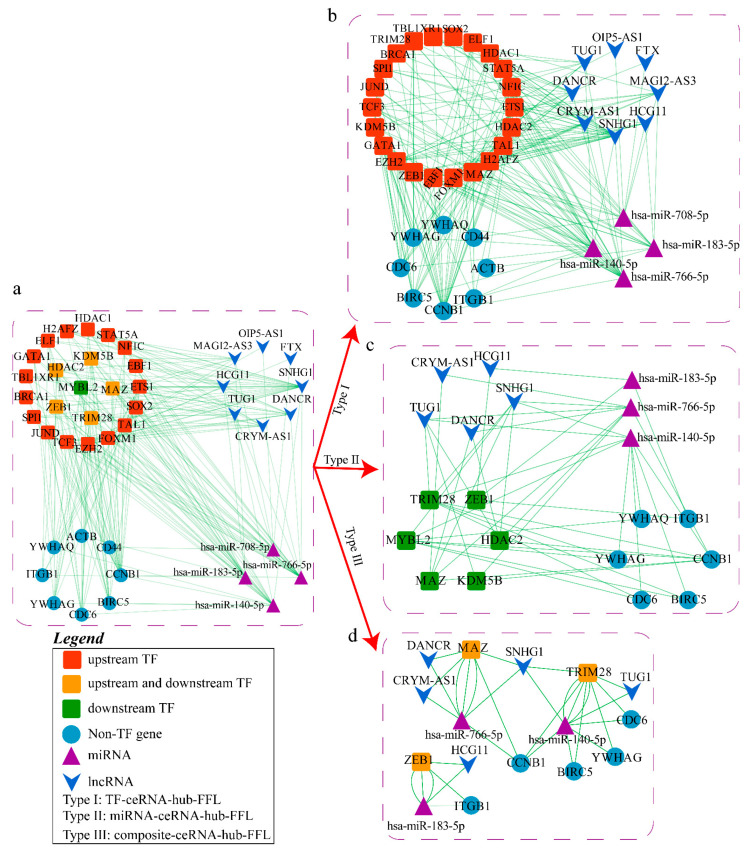
CeRNA-hub-FFL network: (**a**) The total network can be divided into three different types of ceRNA-hub-FFL networks; (**b**) Type I subnetwork; (**c**)Type II subnetwork; (**d**) Type III subnetwork.

**Figure 12 ijms-23-11303-f012:**
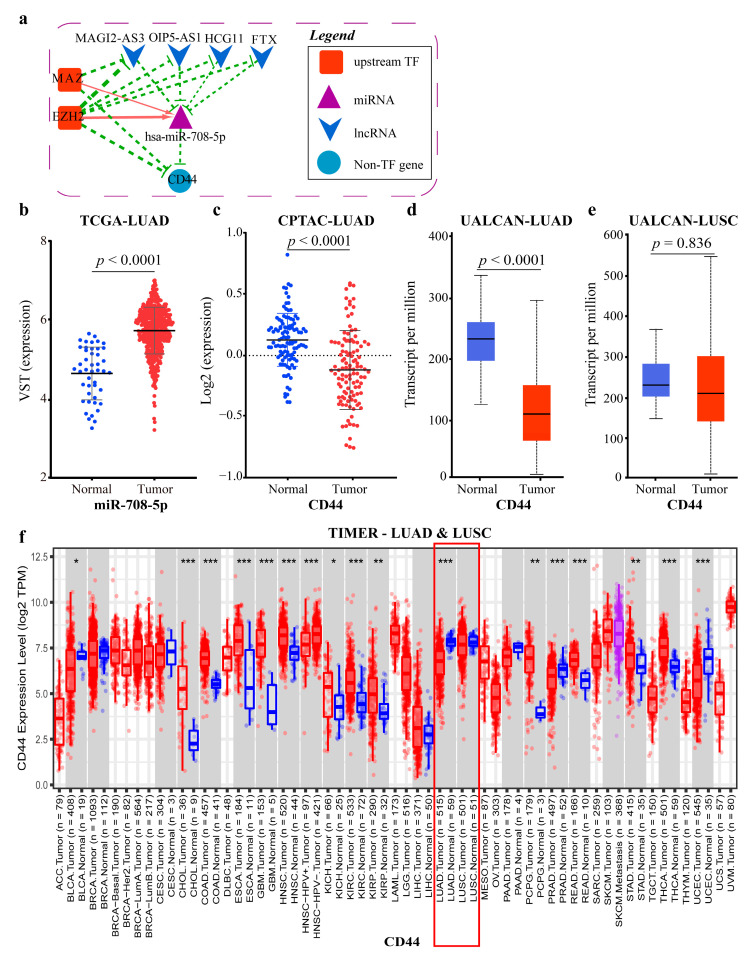
MiR-708-5p related ceRNA-hub-FFL subnetwork and target genes: (**a**) MiR-708-5p related ceRNA-hub-FFL subnetwork (only I-FFL). Green dashed line indicates inhibition, while red indicates activation, and line thickness indicates the strength of correlation; (**b**) MiR-708-5p expression data of LUAD in TCGA; (**c**) *CD44* proteome data of LUAD in CPTAC; (**d**) *CD44* mRNA expression data of LUAD in UALCAN; (**e**) *CD44* mRNA expression data of LUSC in UALCAN; (**f**) *CD44* mRNA expression data of LUAD and LUSC in TIMER. The blue dots/boxs indicate the normal samples, while the red dots/boxs indicate the tumor samples. *: *p* < 0.05; **: *p* < 0.01; ***: *p* < 0.001.

**Figure 13 ijms-23-11303-f013:**
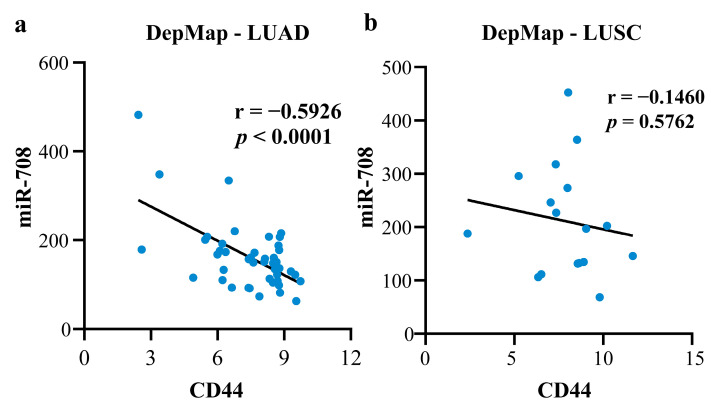
The correlation between miR-708 and *CD44* in NSCLC cell lines: (**a**) The correlation between miR-708 and *CD44* in LUAD cell lines; (**b**) The correlation between miR-708 and *CD44* in LUSC cell lines.

**Figure 14 ijms-23-11303-f014:**
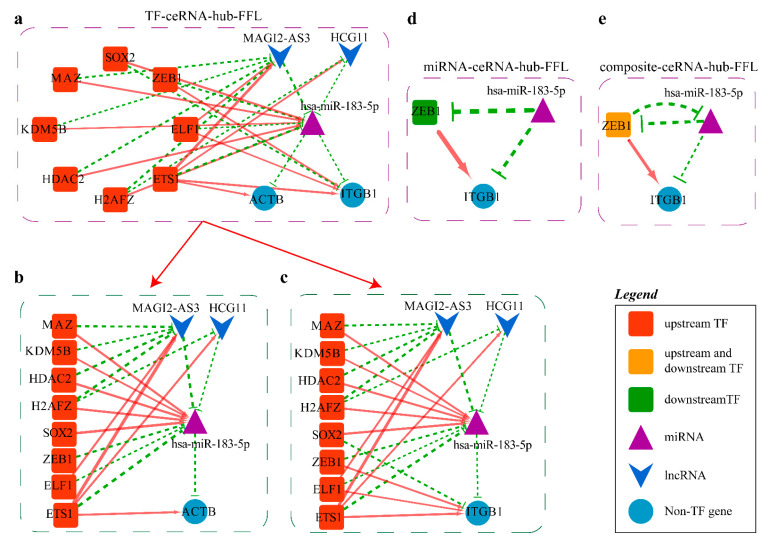
MiR-183-5p related ceRNA-hub-FFL subnetworks: (**a**) MiR-183-5p related I-FFL network; (**b**,**c**) Subplots of individual genes regulated by miR-183-5p in I-FFL; (**d**) MiR-183-5 related II-FFL network; (**e**) MiR-183-5p related III-FFL network. Green dashed line indicates inhibition, red indicates activation, and line thickness indicates the strength of correlation.

**Figure 15 ijms-23-11303-f015:**
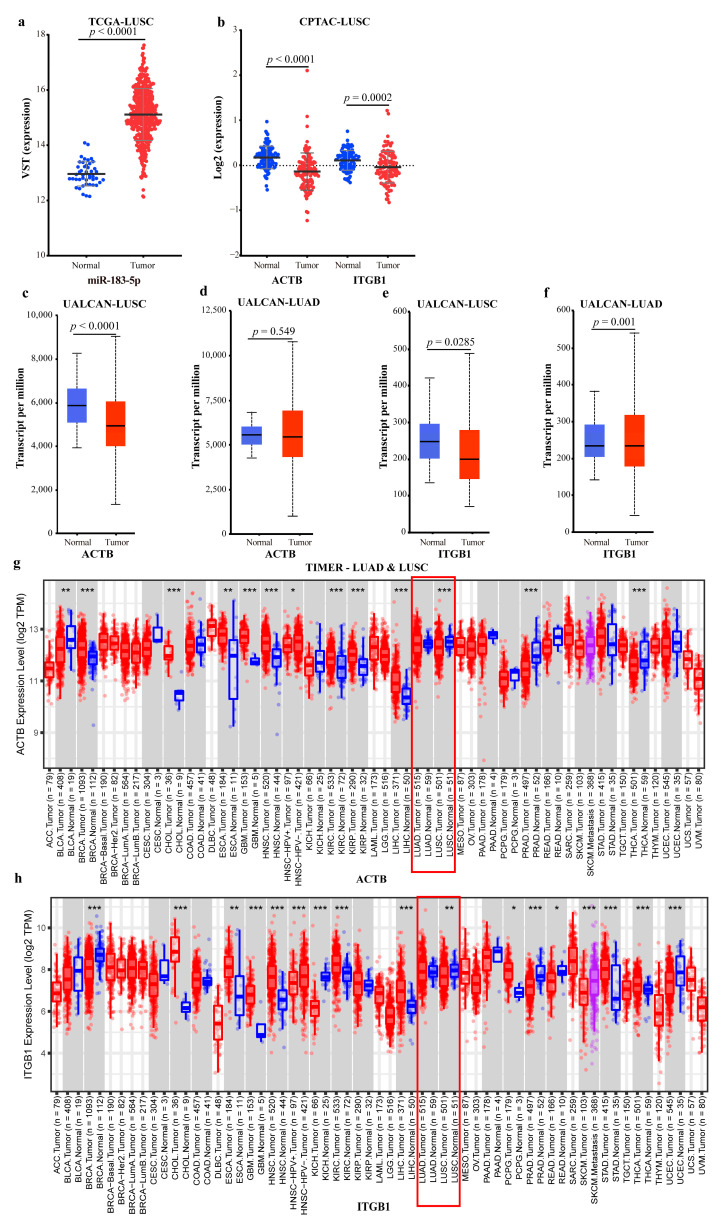
Expression data of miR-183-5p and target genes: (**a**) MiR-183-5p expression data in TCGA; (**b**) Proteome expression data of *ACTB* and *ITGB1* of LUSC in CPTAC; (**c**) *ACTB* mRNA expression data of LUSC in UALCAN; (**d**) *ACTB* mRNA expression data of LUAD in UALCAN; (**e**) *ITGB1* mRNA expression data of LUSC in UALCAN; (**f**) *ITGB1* mRNA expression data of LUAD in UALCAN; (**g**) *ACTB* mRNA expression data of LUAD and LUSC in TIMER; (**h**) *ITGB1* mRNA expression data of LUAD and LUSC in TIMER. The blue dots/boxs indicate the normal samples, while the red dots/boxs indicate the tumor samples. *: *p* < 0.05; **: *p* < 0.01; ***: *p* < 0.001.

**Figure 16 ijms-23-11303-f016:**
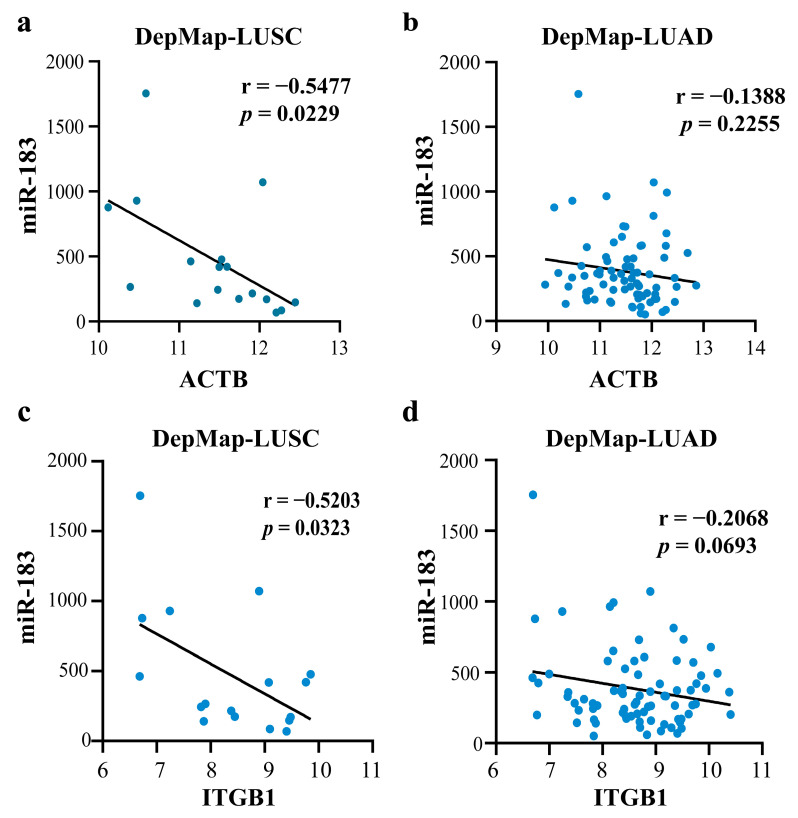
The correlations between miR-183 and *ACTB,* miR-183 and *ITGB1* in NSCLC cell lines: (**a**) The correlation between miR-183 and *ACTB* of LUSC cell lines; (**b**) The correlation between miR-183 and *ACTB* of LUAD cell lines; (**c**) The correlation between miR-183 and *ITGB1* in LUSC cell lines; (**d**) The correlation between miR-183 and *ITGB1* of LUAD cell lines.

**Figure 17 ijms-23-11303-f017:**
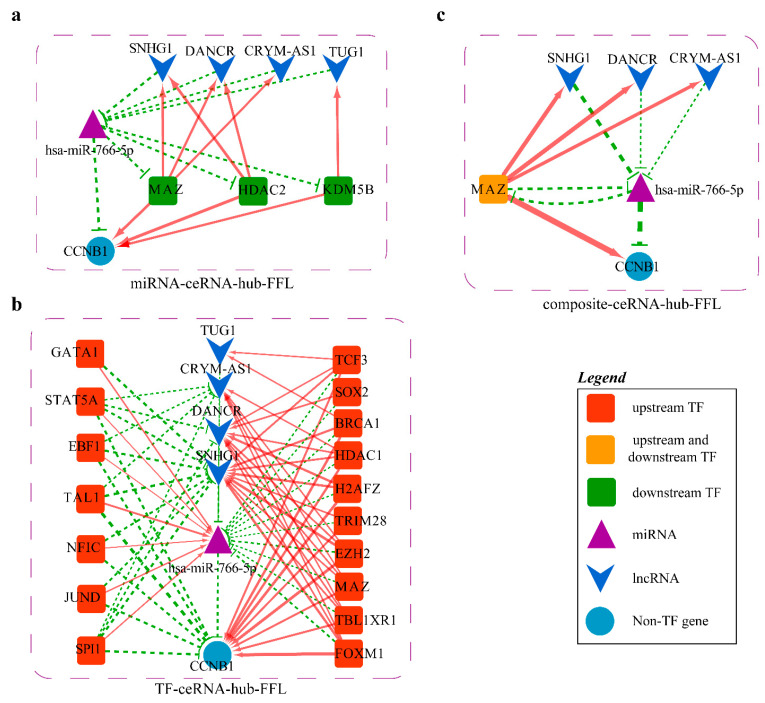
MiR-766-5p related ceRNA-hub-FFL network: (**a**) MiR-766-5p related II-FFL network; (**b**) MR-766-5p related I-FFL network; (**c**) MiR-766-5p related III-FFL network. Green dashed line indicates inhibition, while red indicates activation and line thickness indicates the strength of correlation.

**Figure 18 ijms-23-11303-f018:**
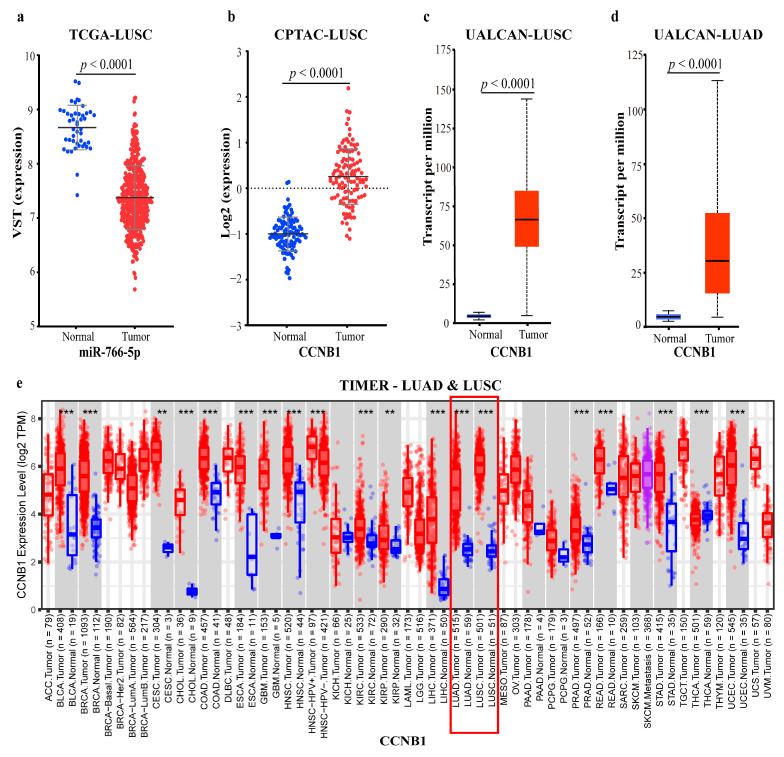
Expression data of miR-766-5p and related gene: (**a**) MiRNA-766-5p expression data of LUSC in TCGA; (**b**) *CCNB1* proteome expression data of LUSC in CPTAC; (**c**) *CCNB1* mRNA expression data of LUSC in UALCAN; (**d**)*CCNB1* mRNA expression data of LUAD in UALCAN; (**e**) *CCNB1* mRNA expression data of LUAD and LUSC in TIMER. The blue dots/boxs indicate the normal samples, while the red dots/boxs indicate the tumor samples. *: *p* < 0.05; **: *p* < 0.01; ***: *p* < 0.001.

**Figure 19 ijms-23-11303-f019:**
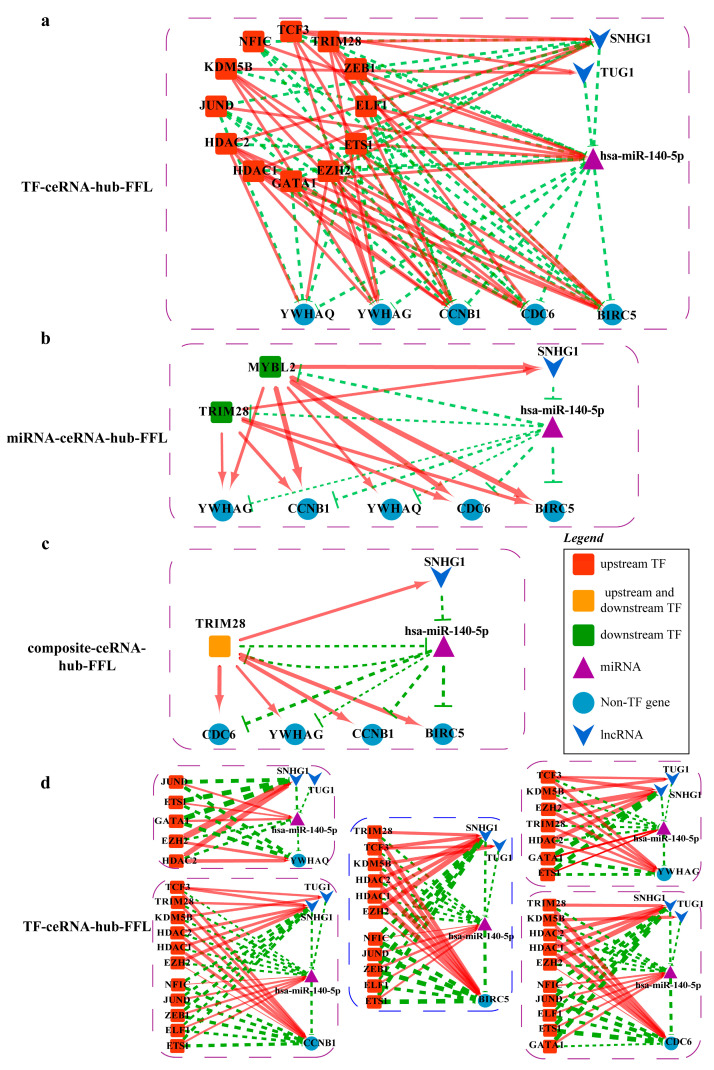
MiR-140-5p related ceRNA-hub-FFL subnetworks: (**a**) MiR-140-5p related I-FFL network; (**b**) MiR-140-5p related II-FFL network; (**c**) MiR-140-5p related III-FFL network; (**d**) Detail map of miR-140-5p related type I ceRNA-hub-FFL. Green dashed line indicates inhibition, while red indicates activation and line thickness indicates the strength of correlation.

**Figure 20 ijms-23-11303-f020:**
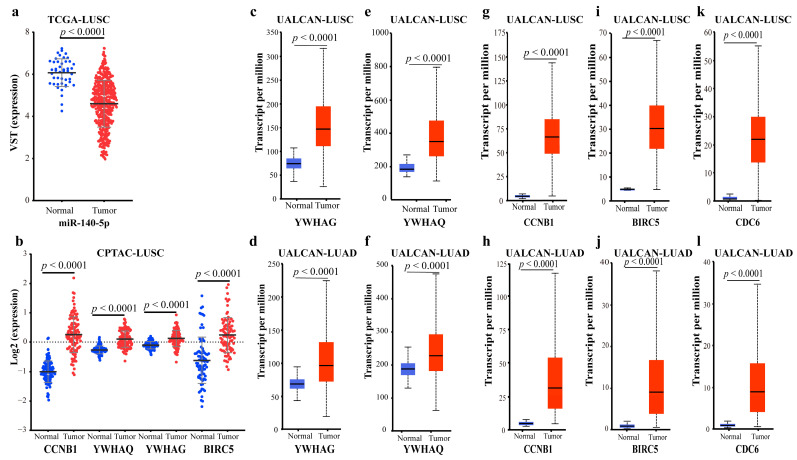
Expression data of miR-140-5p and related proteins: (**a**) MiR-140-5p expression data of LUSC in TCGA; (**b**) Proteome expression data of *CCNB1*, *YWHAQ*, *YWHAG*, and *BIRC5* of LUSC in CPTAC; (**c**) *YWHAG* mRNA expression data of LUSC in UALCAN; (**d**) *YWHAG* mRNA expression data of LUAD in UALCAN; (**e**) *YWHAQ* mRNA expression data of LUSC in UALCAN; (**f**) *YWHAQ* mRNA expression data of LUAD in UALCAN; (**g**) *CCNB1* mRNA expression data of LUSC in UALCAN; (**h**) *CCNB1* mRNA expression data of LUAD in UALCAN; (**i**) *BIRC5* mRNA expression data of LUSC in UALCAN; (**j**) *BIRC5* mRNA expression data of LUAD in UALCAN; (**k**) *CDC6* mRNA expression data of LUSC in UALCAN; (**l**) *CDC6* mRNA expression data of LUAD in UALCAN.

**Figure 21 ijms-23-11303-f021:**
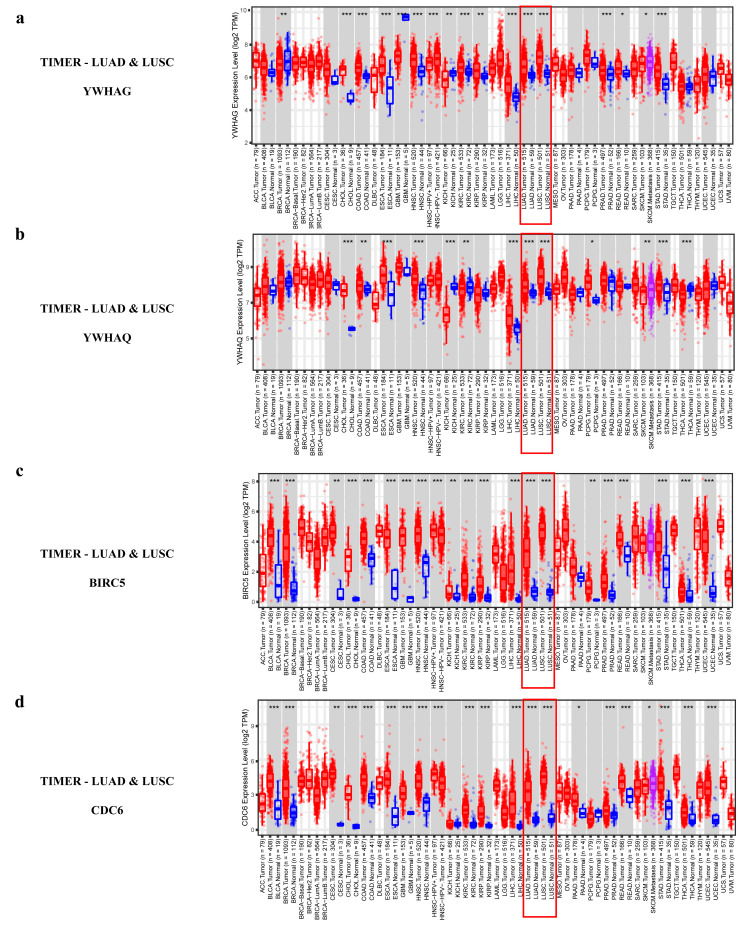
Differential expression of *YWHAG*, *YWHAQ*, *BIRC5*, and *CDC6* in the TIMER: (**a**) *YWHAG* mRNA expression data of LUAD and LUSC in TIMER; (**b**) *YWHAQ* mRNA expression data of LUAD and LUSC in TIMER; (**c**) *BIRC5* mRNA expression data of LUAD and LUSC in TIMER; (**d**) *CDC6* mRNA expression data of LUAD and LUSC in TIMER. The blue dots/boxs indicate the normal samples, while the red dots/boxs indicate the tumor samples. *: *p* < 0.05; **: *p* < 0.01; ***: *p* < 0.001.

**Figure 22 ijms-23-11303-f022:**
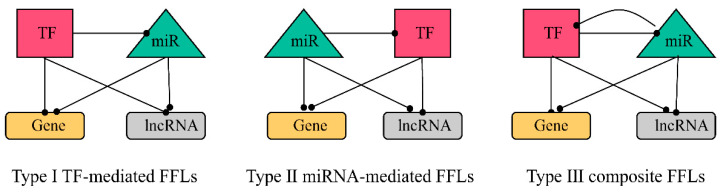
Three types of FFL subnetworks.

**Table 1 ijms-23-11303-t001:** MiRNAs obtained from the intersection of two screening methods in LUAD.

miRNA ID	NOG Value	TFP Value	Important Gene	NOG Gene	AUC
miR-101-3p	8	0.177	EMP1, KLF6	OCIAD2, BRIP1, NPC1L1, KL, FOXF1, AJAP1, NKX6-1, MAOB	0.838
miR-708-5p	6	0.1628	SLIT2	JAM2, TMEM88, MAG, KANK4, DNAH10, FAM107A	0.894

Note The significance test values of ROC curves for all miRNAs: *p*-value < 0.00001.

**Table 2 ijms-23-11303-t002:** MiRNAs obtained from the intersection of two screening methods in LUSC.

miRNA ID	NOG Value	TFP Value	Important Gene	NOG Gene	AUC
miR-101-3p	12	0.1572	FHL1, KLF2	SNX31, CNTN4, PHACTR3, RNASE4, RNASE1, KL, FOXF1, C17orf104, KRT10, CACNA1D, ADD2, MAOB	0.792
miR-140-5p	7	0.2195	BIRC5, CCNB1	TEX19, RASL11B, GRIN1, SRD5A1, FAM162A, ADA, CADPS2	0.874
miR-183-5p	7	0.16	FOS, CAV1, KLF6	E2F8, RAI2, HIST1H2AI, GBP4, DNAH3, FAM83A, FMN1	0.976
miR-766-5p	10	0.1667	CCNB1	EFR3B, NPM3, CALCOCO1, CYP27C1, ASIC1, IL16, PCDHA3, GPBAR1, PPFIA4, KCTD1	0.957
miR-766-3p	16	0.1587	DLC1	SP8, TRIM45, TRIM15, TTBK1, TTC25, IL1RL2, RCCD1, TEKT1, HIST2H2AB, ZNF670, CENPH, TRPV3, CGNL1, VAMP5, KIRREL2, TNNT1	0.973

Note The significance test values of ROC curves for all miRNAs: *p*-value < 0.00001.

**Table 3 ijms-23-11303-t003:** The literature search result of miRNA biomarkers.

miRNA ID	LUAD	LUSC	NSCLC	Subtype Study	Total
miR-708-5p	1	1	6	1	9
miR-766-3p	1	-	1	-	2
miR-766-5p	1	-	-	-	1
miR-183-5p	12	2	19	-	33
miR-140-5p	4	-	19	1	24

## Data Availability

Data is available at NCBI GEO: GSE102286, GSE63805, GSE36681, GSE74190, GSE15008 and TCGA dataset (TCGA-LUAD, TCGA-LUSC, The Cancer Genome Atlas, https://www.cancer.gov, accessed on 18 September 2021).

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
