# Peer review of "A Novel Strategy for Identifying NSCLC MicroRNA Biomarkers and Their Mechanism Analysis Based on a Brand-New CeRNA-Hub-FFL Network"

_ijms, 2022, doi:10.3390/ijms231911303_

Round 1

Reviewer 1 Report

The titled “A Novel Strategy for Identifying NSCLC MicroRNA Bi- omarkers and their Mechanism Analysis Based on a Brand- New CeRNA-Hub-FFL Network” relates to identifying the reliable miRNA biomarkers for the diagnosis of the two major subtypes of NSCLC (LUAD and LUSC). This is a good addition in the area of the subject matter. I suggest the following points to improve the manuscript.

1.     The manuscript has a typo, article usage, and grammatical issues. The authors can rectify these issues with Grammarly software

2.     Some future sentences used in the manuscript must be converted to past tenses, for example, “In this study, we will identify the reliable miRNA biomarkers for the diagnosis of the two major subtypes of NSCLC (LUAD and LUSC), respectively.” It seems that the authors wrote a proposal rather than a manuscript.

3.     A clear objective and conclusion are missing in the abstract.

4.     The conclusion part is also missing in the manuscript.

5.     The introduction part is too long. It may be concise, and some of its components may be used in the discussion.

6.     The impact of the study's findings on the NSCLC treatment needs comprehensive discussion.

7.     The resolution of the figure needs to be increased.

Reviewer 2 Report

The manuscript “A Novel Strategy for Identifying NSCLC MicroRNA Bi-omarkers and their Mechanism Analysis Based on a Brand-New CeRNA-Hub-FFL Network” by Zhang J et al. focused on the identification of new biomarkers in lung cancer as useful tools to predict resistance or favourable therapeutic response in NSCLC.

The article is well structured and the results are clearly presented. However, there are some points that the authors should address to improve the translational relevance of the manuscript: 

-     -The aim of the study is not clearly described in the Abstract and in the Introduction.

-    -The authors should better describe in the Discussion the possible clinical applications of their results

-     - In the abstract Results and Conclusion sections are missing

-       -In the figure 10, the statistical analysis is missing

-      -The authors describe that the miRNAs, that they have identified in NSCLC, may have a controversial role as biomarkers of cancer progression especially when related to the expression of CD44, ACTB and ITGB1 genes. To clarify this point, the authors should assess the link between levels of selected miRNAs and CD44, ACTB and ITGB1 genes expression as well as their oncogene function in different NSCLC cell lines.
